# Exploring the Application of Deep Learning for Supervised Learning Problems

**Jose Rozanec**
Universidad de Buenos Aires

**Gilad Katz, Eui Chul Richard Shin & Dawn Song**
University of California, Berkeley

## Abstract

One of the main difficulties in applying deep neural nets (DNNs) to new domains is the need to explore multiple architectures in order to discover ones that perform well. We analyze a large set of DNNs across multiple domains and derive insights regarding their effectiveness. We also analyze the characteristics of various DNNs and the general effect they may have on performance. Finally, we explore the application of meta-learning to the problem of architecture ranking. We demonstrate that by using topological features and modeling the changes in its weights, biases and activation functions layers of the initial training steps, we are able to rank architectures based on their predicted performance. We consider this work to be a first step in the important and challenging direction of exploring the space of different neural network architectures.

## 1 Introduction

Recent advances in deep neural networks (DNNs) have led to breakthroughs in fields such as image classification (He et al., 2015; Krizhevsky et al., 2012) and speech recognition (Yu et al., 2010; Dahl et al., 2012). One reason for the effectiveness of DNNs is their ability to integrate low, mid and high-level features in a natural way (Zeiler & Fergus, 2014). While recent work such as (Simonyan & Zisserman, 2014) suggests that in many cases the depth of the architecture is crucial, the emergence of more complex architectures (He et al., 2015; Szegedy et al., 2015) demonstrates that depth alone often does not suffice.

While DNNs have been highly effective in several domains, their application in additional fields is yet to become widespread. We argue that this is the case due to two challenges. The first is the difficulty of designing effective architectures for domains in which there is little or no previous knowledge on the application of deep learning. Moreover, since designing DNN architectures is not intuitive for most people, this task is likely to fall to experts whose time is in high demand. The second challenge, which is strongly coupled with the first, is the large amounts of computing power and time required to evaluate multiple DNNs. These traits constrain the number of DNN architectures that can be evaluated, thus further limiting one's ability to explore new architectures or respond to changing circumstances.

In this study we explore the possibility of applying architectures that are effective for one domain to another. We do so by generating a large number of architectures and evaluate their performance on multiple tabular datasets in order to determine whether the architectures are transferable. We also explore the feasibility of architectures with parallel layers and compare their effectiveness to that of their "linear" counterparts. Our results show that while architectures do not perform well across multiple datasets, parallel architectures are surprisingly effective.

When attempting to apply DNNs to an unknown domain, one way of approaching the problem would be to randomly "sample" various architectures and analyze their performance distribution. The top-performing architectures found in the sampling can form the base for future exploration while the variance in performance can assist in determining the number of architectures that need to be sampled. We explore a meta-learning approach that may improve the efficiency of this process by ranking the architectures based on their expected performance. Our approach models the topology of the DNN as well as the changes in weights, biases and activation function layers throughout the initial training steps and uses this information to rank the architectures by their relative performance. Preliminary results are encouraging.

While we consider this study to be an important first step, we feel obliged to point out that work is done in a limited setting. To enable the generation of multiple DNN architectures with diverse topologies, we applied uniform and fixed parameters such as layer sizes and learning rates. As a result, the architecture space we explore is limited. Validating our results on a more diverse set of architectures with multiple hyperparameter configuration will require additional experimentation. We plan to address these issues in future work.

Our contributions are as follows:

- We explore DNNs across multiple datasets, evaluate their effectiveness and analyze if some perform best across datasets.
- We systematically evaluate a large number of architectures over multiple supervised-classification datasets and derive insights regarding the design and application of DNNs with parallel layers for general classification problems.
- We present a novel meta learning-based ranking method that utilizes both topological features as well as weights, biases and activation function layers of the various components of the DNN architecture during the initial training phase. To the best of our knowledge, this is the first time these characteristics have been used in a meta-learning scheme. Preliminary results of this approach are promising.

## 2 RELATED WORK

We review two areas of research whose aim is to better understand and improve the performance of DNN architectures. The first is area of research focuses on the exploration and analysis of DNN architectures. The second area of research is automatic parameter tuning.

### 2.1 EXPLORATION AND ANALYSIS OF DNN ARCHITECTURES

Despite their remarkable success in various domains, the inner-workings of DNNs remain to some degree a "black box". Multiple studies attempted to provide insight into this matter. In Jarrett et al. (2009), the authors analyze convolutional neural networks (CNNs) and derive insights regarding the architecture design and the contribution of its different components. Another work aimed at better understanding CNNs is presented in Shang et al. (2016). The authors analyze widely used CNN architectures and derive insights into their possible shortcomings. To address these shortcomings, they propose a new version of the popular ReLU activation scheme.

The exploration of DNN architectures has also taken place for recurrent neural networks (RNNs). In Zaremba (2015), the authors explore various modifications to LSTM architectures to improve their performance, and propose several enhancements to the architecture. Another study Wu & King (2016) aims to determine the reasons for the effectiveness of LSTMs and identify the contribution of its different elements. Based on their conclusions, the authors proposed a simplified version of LSTM.

### 2.2 AUTOMATIC DNN PARAMETER TUNING

The ability to automatically tune the hyperparameters of a DNN architecture is important not only because of its ability to improve performance, but also due to the considerable time it can potentially save. In Maclaurin et al. (2015) the authors demonstrate how information extracted from the stochastic gradient descent can efficiently tune multiple parameters in the architecture. An additional work that analyzes the gradient is presented in Duvenaud et al. (2016), where the information is used to determine when to terminate the training of the architecture to avoid over-fitting. A different optimization approach is presented in Mendoza et al., where the authors define a large set of hyperparameters (batch size, learning rate, activation types, etc.) and apply Bayesian optimization on top-performing configurations. The approach is only applied to feed-forward networks and outperforms human experts by 10%, using the AUC measure.

Additional types of optimization have also been proposed in recent years. In Jin et al. (2016), the authors focus on setting the size of hidden layers in RNNs. They accomplish this by converting the optimization problem into a subset selection problem. An important aspect of this approach is

that it takes time constraints into account, thus enabling solutions that are feasible given available resources. Another approach, in which one long-short term memory network (LSTM) is used to optimize another, was proposed by Andrychowicz et al. (2016). The two networks have shared parameters but separate hidden states and the optimizer network is both modifying its own weights and those of the optimized simultaneously. Finally, an approach that automatically adjusts the learning rates of the neural net was presented in Schaul et al. (2013). The approach has been shown to be effective both on convex and non-convex learning tasks.

Recent work by Li et al. (2016) proposes an exploration/exploitation scheme for hyperparameter tuning. The authors apply a multi-arm bandits algorithm, with each arm representing a parameter configuration. A process of successive halving (Jamieson & Talwalkar, 2015), in which a certain percentage of the lowest-performing configurations are dropped every $n$ steps enables the framework to explore promising directions. We consider this approach complementary to our proposed meta-learning approach, as the former enables exploration of a large number of configurations while the latter can reduce time required to assess their performance.

## 3 PROBLEM DEFINITION

As mentioned in Section 1, one of the challenges in applying deep learning to a new field is the need to design and test multiple DNN architectures. Only by iterative testing can practitioners discover the capabilities and limitations of deep learning in the domain. Even with ever-increasing computing power, the high computational cost of this process currently presents a significant barrier for most practitioners.

This limitation leads us to explore the following questions:

1. Would DNN architectures that perform well on one general supervised classification problem also be effective when applied to dataset in other domains?
2. What types of architectures are effective for general supervised learning problems? Should practitioners consider other types architectures besides "deep"?
3. Can DNN architectures outperform "conventional" machine learning classifiers in general supervised problems?
4. Is it possible to identify top-performing networks in the early stages of the training? if possible, such a technique could preserve valuable computing resources.

We attempt to begin addressing these questions in the subsequent sections of this study. We iteratively evaluate a large number of DNN architectures on a set of supervised classification problems. These datasets differ from those of image and speech classification in that they consist of tabular data with both numeric and discrete features. These differences make it unclear what types of architectures are likely to perform well on these domains. The datasets we analyze were selected because of their diversity in terms of size and feature number and composition. These traits also enable us to better understand the difficulties in applying DNN architectures across multiple domains.

In order to provide meaningful results, the set of architectures we evaluate is also diverse. We therefore automatically generate a diverse set of architecture with various topological traits. Because little information is available on the application of deep learning to general supervised classification problems, we choose to explore not only architectures that are linear but also architectures with parallel layers. While the generate set is diverse, additional work is required in order to model additional types of architectures. We elaborate on these points further in the subsequent section.

## 4 GENERATING MULTIPLE DNN ARCHITECTURES

In order to effectively explore the architecture space, we require a large and diverse set. We create this set by automatically generating a large number of architectures and training each of them on all training set datasets. Our generation algorithm, presented in Algorithm 1, generates both "deep" and "wide" architectures with parallel layers (see Figure 1(b)). Next we describe the generation process.

We consider DNN architectures to consist of *components*. We define a component as any part of an architecture, be it a layer, normalization or activation function. In this study we consider

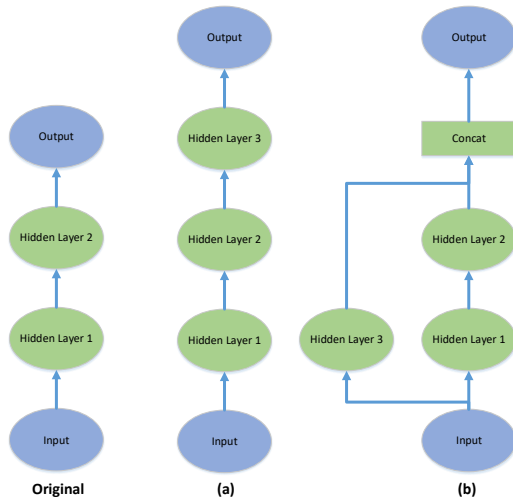

Figure 1: An example of the architectures that can be derived from an existing one.

the following components: fully-connected layers, softmax, batch normalization, dropout and the ReLU, sigmoid and tanh activation functions.

We begin the generation process with a "basic" architecture consisting only of two components: a fully-connected input layer and an output softmax layer. We then expand the set of possible architectures by iteratively applying the following steps:

1. For each pair of components in the architecture, identify all component that could be inserted *between* them (Figure 1(a)).

2. For each pair of components in the architecture, identify all component that could be inserted *in parallel* to one of them (Figure 1(b)).

3. For each of the components identified in the previous steps, generate a new copy of the architecture and perform the corresponding insertion.

Our proposed architecture generation approach enables us to generate the *topological* representation of every possible neural networks that consist of the predefined components. However, we do not generate multiple hyperparameter configurations for each topology and use fixed parameters for each component. We plan to address this limitation in future work, possibly by using an approach similar to the one presented in Li et al. (2016). It is also important to point out that we currently do not support weight-sharing and therefore do not consider CNN and RNN architectures. Given the characteristics of the analyzed data, we do not consider these architecture types likely to produce meaningful results.

Another important aspect of the our architecture generation approach is that we generate architectures with connections between layers of various depths. An example of this is shown in Figure 1(b), where we connect layers of depths 1 and 2. This setting enables us to systematically explore more complex designs than those commonly used. We analyze these architectures further in Section 6.

As the number of possible architectures grows exponentially, we limit the total number of architectures that we generate by constraining the maximal number of components in a architecture and the number of parallel layers an architecture may contain. The specific settings used in our experiments are presented in Section 6.1. These settings were chosen in order to ensure a diverse set of both deep and wide architectures given the time and computing-power constraints, and we plan to change them in future work to further diversify the set of generated architectures. To select the architectures from which additional ones will be generated, we apply a priority queue. We first sort the architectures by the number of their activation layers (in a descending order) with a secondary sorting based on the total number of components (in an ascending order). This setting prioritizes the creation of deeper architectures with multiple activation layers. For each architecture in the final set, we generate the

meta-features described in Section 5. The algorithm for the architecture generation is presented in Algorithm 1.

---

**Algorithm 1** Automatic architecture generation

---
1: **procedure** ARCHITECTUREGENERATION($arcQueue, initArc$)
2: $architecturesSet \leftarrow initArc$
3: $architecturesQueue \leftarrow initArc$
4: **while** ($architecturesQueue \neq \emptyset$) **do**
5: $newarchitectures \leftarrow \emptyset$
6: $architecture \leftarrow arcQueue.pop()$
7: **for each** $P(c_i, c_j) i \neq j \in \{c_1, c_2, ..., c_n\}$ **do**
8: $candidateComponents \leftarrow proposeInsertBetweenCandidates(P(c_i, c_j))$
9: **for each** $candidate \in candidateComponents$ **do**
10: $newarchitecture \leftarrow insertBetween(architecture, P(c_i, c_j), candidate)$
11: $newarchitectures \leftarrow newarchitectures \cup newarchitecture$
12: $candidateComponents \leftarrow proposeInsertAsideCandidates(P(c_i, c_j))$
13: **for each** $candidate \in candidateComponents$ **do**
14: $newarchitecture \leftarrow insertAside(architecture, P(c_i, c_j), candidate)$
15: $newarchitectures \leftarrow newarchitectures \cup newarchitecture$
16: $newarchitectures \leftarrow filter(newarchitectures)$
17: $arcQueue \leftarrow arcQueue \cup newarchitectures$
18: $architecturesSet \leftarrow architecturesSet \cup newarchitectures$
19: **return** $architecturesSet$

---

## 5 META-LEARNING FOR ARCHITECTURE RANKING

Our goal is to determine whether by analyzing the topology of DNN architecture as well as the transformations it undergoes in its early training iterations could be used to predict its performance. To this end we develop a novel machine learning-based approach that generates a set of *features* for each analyzed architecture. Once the features are generated, we use a ranking classifier to assign a score to each architecture. The classifier is trained on a large corpus of datasets (additional information is provided in Section 6.1).

We apply meta-learning (Vilalta & Drissi, 2002) to predict the performance of the DNN architectures. Meta-learning is a branch of machine learning in which an algorithm "learns how to learn" by extracting information on the learning process of another algorithm. The features extracted in this process are called meta-features. We generate three types of meta-features: *dataset-based*, *topology-based* and *training-based*. We hypothesize that these groups represent the elements that affect the performance of the DNN architecture - the data on which it is trained, the structure of the network and the changes in its weights, biases and activation functions during throughout the training process. We provide a full overview of the meta-features groups below and detailed information in Appendix A .

**Dataset-based meta-features.** As explained in Section 3, the datasets we use in the evaluation vary significantly in size and feature composition. These meta-features attempt to represent the multiple characteristics that may affect the performance of deep learning algorithms. We generate three types of meta-features:

1. **General information:** general statistics on the analyzed dataset: number of instances and classes, number and type of features and statistics on the correlations among various features.

2. **Entropy-based measures:** we partition the dataset's features based on their type (discrete, numeric, etc.) and calculate statistics on the Information Gain (IG) of the features in each group.

3. **Feature diversity:** we partition the dataset into type-based groups and use the chi-squared and paired-t test to calculate the similarity of each pair in each group. We then generate meta-features using the tests' statistic values.

**Topology-based meta-features.**   Our generated architectures vary significantly in size, depth and width. Since these traits are likely to affect their performance, we use the meta-features of this group to quantify and model them. The meta-features can be partitioned into two groups:

1. **Architecture composition:** general statistics on the number and types of layers and functions that make up the architecture, statistics on layer composition as a function of depth etc.

2. **Connectivity-based measures:** for each layer in the architectures, we calculate various measures that are frequently used for graph-analysis. These measures include statistics on the number and ratio of incoming and outgoing edges (overall, per depth and per type) and node-centrality evaluation measures.

**Training-based meta-features.**   The goal of these meta-features is to model the transformations undergone by the DNN during the course of its training. These meta-features consist of statistics on the weights, biases and activation function layers of the various components in the architecture. These meta-features can be partitioned into two groups:

1. **Static evaluation:** general statistics on the distribution of the various values across different depths and layer types. These features provide "snapshot" information on the training status of the architecture in multiple training steps.

2. **Time series-based evaluation:** We compare the values obtained in the various training iterations to those obtained earlier, calculate ratios and modeling the changes in values distribution over time.

A full description of all meta-features is provided in Appendix A.

# 6   EXPERIMENTS AND ANALYSIS

## 6.1   EXPERIMENTAL SETUP

We conduct our experiments on 13 supervised classification datasets in a tabular form. We selected these datasets since they represent common supervised-learning problems that are not often addressed by deep learning. In addition, their feature composition consists of both numeric and discrete features, a trait that makes them different from image and speech classification datasets. The datasets vary significantly in size, number and type of features (some contain only numerical features while others also contain discrete features) and class imbalance - traits we hypothesize will make learning across domains more challenging. All datasets are available on the OpenML repository and their properties are represented in Appendix B.

We use the following settings:

- For each dataset, we train *the same set* of 11,170 architectures, generated as described in Section 4. The maximal width (number of parallel layers) allowed for an architecture was set to 4, and we terminated the generation process upon reaching the predefined number of architectures. This deepest architectures generated by this approach have 8 activation layers and 14 components overall.

- For architectures training, all datasets were randomly partitioned into training, validation and test sets. 80% of the data points was used by the training and the remaining two sets assigned 10% each. The same split was used for all the architectures explored for each dataset. Original class ratios were maintained in all sets.

- All generated architectures were trained until convergence, with the time of termination determined by performance on the validation set.

- The training-based meta-features were only extracted for the following steps: 20, 40, 60, 80 and 100.

- We used a leave-one-out (LOO) cross-validation approach to train the ranking classifier: for each evaluated dataset $d_i$, we train the ranking classifier using the meta-features from $d_j \in D$ where $i \neq j$. This setting enables to test whether a meta-model trained on one dataset could be effectively applied on another.

- We randomly split the generated architectures into two groups. The first group, consisting of 70% of the architectures, is used for training. We use the remaining 30% to evaluate the performance of our approach on each dataset.

## 6.2 ANALYSIS

We begin by analyzing the accuracy distribution of the generated architectures across the datasets. We found that the distribution of accuracies varies significantly across the different datasets, with some datasets with ranges of [45%-90%] accuracy while others are in the range [89%-95%]. This difference has significant impact on one's ability to apply architectures that are effective in one domain to another, as we confirm with the next experiment. An example of accuracies distributions is presented in figures 2 and 3 and plots for all datasets are presented in Appendix D.

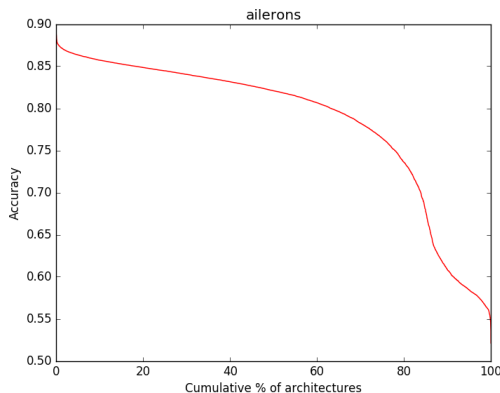 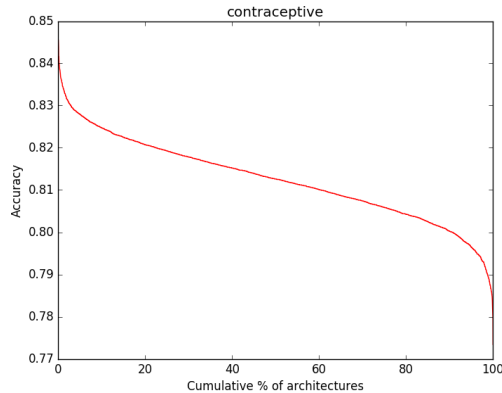

Figure 2: Accuracies plot for the dataset Ailerons

Figure 3: Accuracies plot for the dataset Contraceptive

**Analyzing the performance differences of "parent–child" architectures.** In order to determine whether our architecture generation method is effective, we analyzed the differences in accuracy between every architecture and its descendant. Our reason for performing this analysis is as follows: if making incremental additions to an existing architecture does not significantly change its performance, then we simply generate a large number of architecture which are nearly identical in performance.

The results of our analysis are presented in Table 1. For every "parent–child" pair we calculate the difference in accuracy on the test set. We then calculate the maximal and average changes in accuracy for each dataset. It is clear from the results that the changes in accuracy are significant, especially given the fact that changes are accumulated over time (deeper architectures are a result of multiple modifications).

Next we analyze the "parent–child" architectures with the maximal differences in order to determine whether the addition of particular component is most likely to induce large changes in accuracy. Our results, presented in Table 2, show that no one component type can be consistently attributed with inducing large changes.

**Applying architectures across datasets.** We attempt to determine whether it is possible to find architectures that perform well across multiple datasets. For each of the generated architectures, we calculate its performance-based ranking (i.e. position in a list ordered by the accuracy measure) on each of the datasets. Then, for each dataset we test the performance of the architecture with the *best average ranking* on the remaining datasets. We compare the performance of this architecture to that of the best evaluated architecture and to that of the best architecture found by our meta-learning model (described in the following section). The results, presented in Table 3, show significant differences in performance and lead us to conclude that in most cases DNN architectures do not perform well across multiple datasets.

Table 1: Analyzing the differences in accuracy for the different architecture parent–child pairs for each dataset.

| Dataset | Max difference | Average difference |
|---|---|---|
| Contraceptive | 5% | 1.8% |
| Seismic bumps | 4.9% | 1.1% |
| Page Blocks | 7.4% | 1.4% |
| Wind | 35% | 3.2% |
| Puma_32 | 19.2% | 1.8% |
| CPU_act | 40% | 3.3% |
| Delta elevators | 39.5% | 2.7% |
| Mammography | 3% | 1.1% |
| Ailerons | 17.4% | 5.7% |
| Bank marketing | 3.5% | 0.8% |
| German Credit | 5% | 1% |
| Space | 11.5% | 2.5% |
| Cardiography | 11.5% | 1% |

Table 2: Analyzing the differences in accuracy for the different architecture parent–child pairs for each dataset.

| Component type | Number of appearances |
|---|---|
| Dropout | 2 |
| Sigmoid | 3 |
| TanH | 2 |
| Fully connected | 2 |
| ReLU | 1 |
| Batchnorm | 3 |

**Comparing the performance of DNN architectures to those of "conventional classifiers".**    As a point of reference to "classical" machine learning approaches for classifying tabular data, in Table 3 we also presents the performance of the Random Forest algorithm (using the Weka Hall et al. (2009) implementation with the default parameters ). It is clear that neither Random Forest nor the DNN architectures consistently outperform the other. We intend to explore the factors that cause these differences in performance in future work.

Table 3: Comparison of the accuracy performance of the best average-ranking architectures to the top-ranking architecture found by our approach for each dataset.

| Dataset | Best architecture | Top ranked (best found by model) | Architecture with best average ranking | Random Forest |
|---|---|---|---|---|
| Contraceptive | 84.5% | 84% | 79.7% | 76.4% |
| Seismic bumps | 95% | 94.1% | 92.1% | 93.4% |
| Page Blocks | 97% | 95.2% | 89.6% | 97.9% |
| Wind | 88% | 84.3% | 54% | 86.5% |
| Puma_32 | 70% | 67% | 50.7% | 88.1% |
| CPU_act | 91% | 87.7% | 70% | 93.7% |
| Delta elevators | 90% | 88.7% | 79.2% | 87.7% |
| Mammography | 99% | 98.9% | 97% | 98.8% |
| Ailerons | 89% | 86.2% | 59% | 88.6% |
| Bank marketing | 96% | 95% | 94% | 90.5% |
| German Credit | 77.1% | 73.6% | 68.2% | 76.9% |
| Space | 69.6% | 66.8% | 56.5% | 84% |
| Cardiography | 94.5% | 93.7 | 86.4% | 95.5% |

**Analyzing the performance of architectures with parallel layers.**    Next we explore whether architectures with parallel layers outperform similar non-parallel architectures. We analyze the 100 top-performing architectures of each dataset and calculate the percentage of architectures with parallel layers. The results, presented in Appendix C, show that this type of architecture consists on average of 62% of the top-performing architectures.

To determine whether the benefit of applying parallel layers is significant, we randomly choose one of our datasets (Ailerons) and identify the 100 top-performing architectures with parallel layers. From this set we randomly sample 10 architectures and compare the performance of each of them to those of *all* of their possible serial counterparts, created by iteratively removing all but one of the different parallel layers. Our results, presented in Table 4, show that architectures with parallel layers significantly outperform *all* of their serial counterparts.

Considering the same sample of parallel architectures, we analyze whether architectures performance can be improved by adding a batch normalization before, after or before and after each activation function. As shown by the results in Table 4, we did not find evidence that the addition of batch normalization improves the performance of architectures with parallel layers. We find this fact surprising and intend to explore this further in future work. An example of one of the parallel architectures is presented in Figure 4 in Appendix C.

Finally, we also analyze the component composition of the 100 top-performing architectures for each dataset. The most interesting conclusion found in this analysis is the fact that a relatively shallow architectures ( 4 fully-connected layers) seem to yield the best performance on average for all datasets. The full analysis of the architecture components is presented in Table 12 in Appendix C.

Table 4: Comparison of the performance of parallel architectures to their serial counterparts.

|  | Parallel Architectures | Serial versions | Parallel with batchnorm – before | Parallel with batchnorm – after | Parallel with batchnorm – before & after) |
|---|---|---|---|---|---|
| Average | 87.6% | 71.8% | 70.4% | 77.4% | 76.5% |
| Standard Deviation | 0.39% | 7.8% | 9.9% | 4.2% | 3.6% |

## 6.3 EVALUATING THE META-LEARNING APPRAOCH

We analyze the performance of our meta-learning model as a classifier to rank architectures based on their performance. For these experiments, we use the following settings:

- We define the 5% of the top-performing architectures of each dataset as "good" and label the remaining as "bad". We use this setting due to the large variance in the performance of the DNN architectures on the different datasets (see Appendix D for full details). We also intend to experiment with other labeling methods in future work.

- We use the precision@X measure as the evaluation metric. We calculate it by ranking all architectures according with the confidence of the meta-classifier (i.e. the classifier trained on the meta-features) in them being "good". Then, for the $X$ top-ranking architectures we calculate the actual percentage of "good" architectures is $X$.

- We conduct a separate evaluation on the training-based meta-features and the dataset-based and topological meta-features. Since the training-based features are more computationally expensive to compute, we find it interesting to compare their performance to the other types of meta-features. In our experiments we denote the full set as $ML_{full}$, the training-based meta-features as $ML_{train}$ and the topological and dataset-based meta-features as $ML_{data+top}$.

- We use the Random Forest algorithm for the training of the meta-model.

The results of our evaluation are presented in Table 5. We show that we are able to identify multiple architectures in the top-ranking spots in a much higher ratio than their share of the population. It is also clear that the joint set of all meta-features outperforms both of the examined subsets.

Next we conduct a random sampling over architectures, and compare the performance of the sampled architectures to those obtained by ranking all architectures using the proposed meta-classifier. Our goal is to determine the probability that $N$ randomly-sampled architectures will consist of at least one architecture that outperforms all the top $M$ items ranked by the meta-classifier. We conduct the experiment as follows: for each dataset, we randomly sample a fixed number of architectures and identify the one with the highest performance among those sampled. We then check if this

architecture outperforms all those in the ranked list provided by the meta-learning model. We repeat this process 50,000 for each dataset and calculate the probability of this scenario. The results, presented in Table 6, show that our model outperforms random sampling for all datasets, often by a large margin. However, further experimentation is required to fully determine the effectiveness of the meta-learning approach.

Finally, we analyze the results in order to determine the effectiveness of the different meta-features used by our model. The analysis was carried out by running LASSO logistic regression and analyzing the weights assigned to the various meta-features. Based on this analysis we reach the following conclusions:

- The dataset-based meta-features had the smallest contribution to the performance. While it is somewhat surprising given the fact that DNNs perform very differently on dataset with different characteristics, we conclude that the model is focused on the way in the architecture is trained on the data (i.e. weights and activations).

- The topological meta-features that had the largest contribution were those modeling the depth of the network, the number of parallel layers and those counting the number of various components.

- The ranking model uses a large number of training-based meta-features and from all types described in Appendix A. However, the model includes only weight and activation-based meta-features among the training-based meta-features. The biases-based meta-features are almost never used.

Table 5: The evaluation results of different approaches using the precision@X metric. $full$, $train$ and $d+t$ denote $ML_{full}$ (all meta-features), $ML_{train}$ (training-based meta-features only) and $ML_{data+top}$ (dataset-based and topological meta-features) respectively. Best results are in bold.

| Dataset | precision@5 | | | precision@10 | | | precision@20 | | | precision@50 | | |
|---|---|---|---|---|---|---|---|---|---|---|---|---|
| | $full$ | $train$ | $d+t$ | $full$ | $train$ | $d+t$ | $full$ | $train$ | $d+t$ | $full$ | $train$ | $d+t$ |
| Contraceptive | **20%** | **20%** | 0% | **20%** | 10% | **20%** | **20%** | 5% | 15% | **20%** | 10% | 8% |
| Seismic Bumps | 20% | **40%** | 20% | **20%** | **20%** | 10% | **25%** | 20% | 15% | 12% | **16%** | 12% |
| Page Blocks | **40%** | 20% | 0% | **30%** | 20% | 0% | **20%** | 15% | 0% | **16%** | 14% | 14% |
| Wind | **40%** | 0% | **40%** | 20% | 20% | **30%** | 10% | 15% | **25%** | 12% | 16% | **20%** |
| Puma32 | **20%** | **20%** | 0% | 10% | **20%** | **20%** | 15% | **20%** | 10% | **16%** | 10% | 10% |
| CPU_Act | **40%** | 20% | 20% | **30%** | 20% | 20% | **30%** | 15% | 10% | **22%** | 12% | 16% |
| Delta Elevators | **20%** | **20%** | **20%** | **20%** | **20%** | 10% | 15% | **25%** | 20% | **20%** | **20%** | 12% |
| Mammography | **20%** | 0% | 0% | **20%** | **20%** | 0% | **20%** | 15% | 5% | **20%** | 10% | 12% |
| Ailerons | **40%** | **40%** | **40%** | **30%** | **30%** | 20% | **30%** | 20% | 20% | **28%** | 22% | 26% |
| Bank Marketing | **20%** | 0% | 20% | **30%** | 10% | 20% | **20%** | 10% | 10% | 10% | **14%** | 10% |
| German Credit | **40%** | 20% | 20% | **40%** | 10% | 10% | **20%** | 10% | 10% | **14%** | 10% | 10% |
| Space | **20%** | 0% | 0% | **10%** | 10% | 0% | **15%** | 10% | 10% | **18%** | 14% | 10% |
| Cardiography | **20%** | 0% | **20%** | **20%** | 10% | 10% | **20%** | 15% | 20% | **18%** | 14% | 16% |

## 7 CONCLUSIONS AND FUTURE WORK

In this study we have explored several aspects of applying DNNs to supervised classification problems. Our results demonstrate the difficulty in using DNN architectures that are effective in one domain to another. We also systematically compare the performance of architectures with parallel layers to those of similar linear architectures and demonstrate that the former outperforms the latter in many cases. We present a novel approach for predicting the performance of a DNN architecture by analyzing its topology and the changes in its weights, biases and activation function values during early phases of training. Our aim is that this work can lay the foundation for a better understanding of the DNN architectures space.

For future work we consider several directions. First, we plan to add additional components to the ones currently used in our automatic architecture generation method in order to enable further exploration. In addition, we will seek to enhance our approach adding automatic parameter tuning methods. This will enable us to efficiently explore multiple configurations and possibly identify higher-performing architectures. We are also considering the use of an exploration/exploitation

Table 6: The probabilities of finding an architecture that outperforms all those in the ranked list when randomly sampling a set of architectures. The size of the ranked list by our algorithm is always 10 (i.e. for sample size 20 we test a set two times the size of the ranked list.)

| Dataset | Sample size - 10 | Sample size - 20 |
|---|---|---|
| Contraceptive | 1.7% | 3.2% |
| Seismic bumps | 11.5 | 22% |
| Page Blocks | 14.8% | 27.7% |
| Wind | 24.3% | 41.5% |
| Puma_32 | 20.7% | 36.5% |
| CPU_act | 3.4% | 6.7% |
| Delta elevators | 33.3% | 55.5% |
| Mammography | 7.5% | 14.3% |
| Ailerons | 13.9% | 25.5% |
| Bank marketing | 5.6% | 10.4% |
| German Credit | 11.9% | 22.9% |
| Space | 20.2% | 36.3% |
| Cardiography | 5.6% | 11.2% |

scheme along the lines presented in Li et al. (2016) to enable us to efficiently explore larger architecture spaces.

Another approach we plan to explore is to make the search over network architectures a fully-differentiable problem, by encoding the problem only using mechanisms that enable such a search. As an example, let us imagine that we want to decide the best number of internal hidden layers to use in a multi-layer fully-connected neural net. For this, we could create multiple parallel stacks of layers with the same input at the bottom (e.g. the features for each data point) and the same kind of output at the end (e.g. probabilities over the possible classes) and then use a softmax to take a weighted sum of the outputs from each of the parallel stacks. By using a penalty on the negative entropy of this weighted sum, and increasing the penalty over time, the network should learn to produce the output using only one of the parallel stacks which we can then use at inference time. We can also train multiple models simultaneously using this method, and introduce additional penalties to ensure that the multiple models explore different architectures during training, to enable a more diverse search.

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

## A   THE META-FEATURES USED BY OUR APPROACH

We extract three types of meta-features for the training: dataset-based, topological and training-based. We now provide a complete description of the features used. Note that because calculate the meta-features in Table 9 three times: for the weights, biases and activation functions.

Table 7: Description of the dataset-based meta-features used by our approach.

| Features Name | Description |
|---|---|
| numOfInstances | The number of instances in the dataset. |
| numOfClasses | The number of classes in the dataset. |
| numOfFeatures | The number of features in the dataset. |
| numOfNumericFeatures | The number of numeric (continuous) features in the dataset. |
| numOfDiscreteFeatures | The number of discrete (non-numeric) features in the dataset. |
| ratioNumericFeatures | The percentage of the numeric features of all features. |
| ratioDiscreteFeatures | The percentage of the discrete features of all features. |
| {max,min, avg, stdev}DiscFeatVals | Statistics on the number of possible values for a discrete feature. |
| {max,min, avg, stdev}IGVal | For every feature we calculate the information gain. We then generate statistics on the set of values. |
| {max,min, avg, stdev}NumericIGVal | Same as the previous meta-feature, but calculated only for numeric features. |
| {max,min, avg, stdev}DiscreteIGVal | Same as the previous meta-feature, but calculated only for discrete features. |
| {max,min, avg, stdev}PairedTT | For every pair of numeric features we calculate the statistic of a paired-t test. We then generate statistics on the values. |
| {max,min, avg, stdev}ChiSquareAll | For every pair of features we calculate the statistic of a Chi-Square test. We then generate statistics on the values. |
| {max,min, avg, stdev}ChiSquareDisc | For every pair of discrete features we calculate the statistic of a Chi-Square test. We then generate statistics on the values. |

Table 8: Description of the topological meta-features used by our approach.

| Features Name | Description |
|---|---|
| numOfVertices | The number of vertices in the architecture |
| numOfEdges | The number of edges in the architecture |
| {max,min,avg,stdev}IncomingEdges | Statistics on the number of incoming edges, calculated over all components. |
| {max,min,avg,stdev}OutgoingEdges | Statistics on the number of outgoing edges, calculated over all components. |
| {max,min,avg,stdev}DepthsPerVertex | Because of the parallel layers, a vertex may have multiple depths. We calculate statistics on these values across all components. |
| {max,min,avg,stdev}VerticesPerDepth | For each depth in the architecture, we count the number of components that are in the said depth. We then calculate statistics across all depths. |
| {max,min,avg,stdev}Betweenness | For every component in the architecture, we calculate its betweenness centrality measure. We then calculate statistics across all components. |
| {max,min,avg,stdev}BetweennessNorm | Same as the previous set of meta-features, but the betweenness values are normalized. |
| {num,ratio}MaxPool | The number of MaxPool Layers in the architecture and their ratio of the overall number of components. |
| {num,ratio}Concat | The number of Concatination layers in the architecture and their ratio of the overall number of components. |
| {max,min,avg,stdev}FCSize | Statistics on the size of the fully-connected layers in the architecture. |

Table 9: Description of the training-based meta-features used by our approach.

| Features Name | Description |
|---|---|
| {max,min,avg,stdev}GlobalMax | For every component, get the maximal value of the analyzed trait. We then calculate statistics for all components |
| {max,min,avg,stdev}GlobalMaxRatio | For each meta-feature in the previous line, divide its value in the value of same meta-feature calculated at initialization |
| {max,min,avg,stdev}GlobalMin | For every component, get the minimal value of the analyzed trait. We then calculate statistics for all components |
| {max,min,avg,stdev}GlobalMinRatio | For each meta-feature in the previous line, divide its value in the value of same meta-feature calculated at initialization |
| {max,min,avg,stdev}GlobalAvg | For every component, get the average of the values of the analyzed trait. We then calculate statistics for all components |
| {max,min,avg,stdev}GlobalAvgRatio | For each meta-feature in the previous line, divide its value in the value of same meta-feature calculated at initialization |
| {max,min,avg,stdev}GlobelStdev | For every component, get the standard deviation of the values of the analyzed trait. We then calculate statistics for all components |
| {max,min,avg,stdev}GlobalStdevRatio | For each meta-feature in the previous line, divide its value in the value of same meta-feature calculated at initialization |
| {max,min,avg,stdev}ByTypeMax | For each type of components, get the maximal value of the analyzed trait. We generate separate meta-features for each component type (i.e. multiple sets of features are generated). |
| {max,min,avg,stdev}ByTypeMaxRatio | For each meta-feature in the previous line, divide its value in the value of same meta-feature calculated at initialization |
| {max,min,avg,stdev}ByTypeMin | For each type of components, get the minimal value of the analyzed trait. We generate separate meta-features (i.e. multiple sets of features are generated). |
| {max,min,avg,stdev}ByTypeMinRatio | For each meta-feature in the previous line, divide its value in the value of same meta-feature calculated at initialization |
| {max,min,avg,stdev}ByTypeAvg | For each type of components, get the average of values of the analyzed trait. We generate separate meta-features (i.e. multiple sets of features are generated). |
| {max,min,avg,stdev}ByTypeAvgRatio | For each meta-feature in the previous line, divide its value in the value of same meta-feature calculated at initialization |
| {max,min,avg,stdev}ByTypeStdev | For each type of components, get the standard deviation of the values of the analyzed trait. We generate separate meta-features (i.e. multiple sets of features are generated). |
| {max,min,avg,stdev}ByTypeStdevRatio | For each meta-feature in the previous line, divide its value in the value of same meta-feature calculated at initialization |
| {max,min,avg,stdev}ByDepthMax | For all components at a given depth, identify the maximal value. Then, generate the statistics across all depths. |
| {max,min,avg,stdev}ByDepthMaxRatio | For each meta-feature in the previous line, divide its value in the value of same meta-feature calculated at initialization |
| {max,min,avg,stdev}ByDepthMin | For all components at a given depth, identify the minimal value. Then, generate the statistics across all depths. |
| {max,min,avg,stdev}ByDepthMinRatio | For each meta-feature in the previous line, divide its value in the value of same meta-feature calculated at initialization |
| {max,min,avg,stdev}ByDepthAvg | For all components at a given depth, identify the average value. Then, generate the statistics across all depths. |
| {max,min,avg,stdev}ByDepthAvgRatio | For each meta-feature in the previous line, divide its value in the value of same meta-feature calculated at initialization |
| {max,min,avg,stdev}ByDepthStdev | For all components at a given depth, identify the standard deviation value. Then, generate the statistics across all depths. |
| {max,min,avg,stdev}ByDepthStdevRatio | For each meta-feature in the previous line, divide its value in the value of same meta-feature calculated at initialization |

## B   Full information on the datasets used in the evaluation

Table 10: The characteristics of the datasets used in the experiments

| Name | Num of Data Points | % of Minority Class | Num of Features | % of Numeric Features |
|---|---|---|---|---|
| German Credit | 1,000 | 30% | 20 | 30% |
| Contraceptive | 1,473 | 22.6% | 9 | 66.6% |
| Cardiography | 2,126 | 22.1% | 22 | 100% |
| Seismic bumps | 2,584 | 6.5% | 18 | 77% |
| Space | 3,107 | 49.5% | 6 | 100% |
| Page Blocks | 5,473 | 9.3% | 10 | 100% |
| Wind | 6,574 | 46.7% | 14 | 100% |
| Puma_32 | 8,192 | 49.6% | 32 | 100% |
| CPU_act | 8,192 | 30.2% | 21 | 100% |
| Delta elevators | 9,517 | 49.7% | 6 | 100% |
| Mammography | 11,183 | 2.3% | 6 | 100% |
| Ailerons | 13,750 | 42.3% | 40 | 100% |
| Bank marketing | 45211 | 11.6% | 16 | 43.75% |

## C   Analysis of the performance parallel layers

For each dataset, we analyze the 100 top-performing architectures and determine the percentage of architectures with parallel layers. The results, presented in Table 11, show that the percentage is significant. In table 12 we analyze the component composition of these architectures. The most interesting point (in our view) is that the number of fully-connected layers is about half of the possible maximum. We take this as an indication that the creation of very deep DNNs may not be required for tabular datasets of the type analyzed in this work. In Figure 4 we present an example of an architecture with parallel layers that was among the 100 top-performing on the Ailerons dataset.

Table 11: The percentage of architectures with parallel layers in the 100 top-performing architectures for each dataset.

| Dataset | % of architectures with parallel layers |
|---|---|
| Contraceptive | 61% |
| Seismic bumps | 60% |
| Page Blocks | 65% |
| Wind | 61% |
| Puma_32 | 59% |
| CPU_act | 64% |
| Delta elevators | 73% |
| Mammography | 61% |
| Ailerons | 61% |
| Bank marketing | 62% |
| German Credit | 59% |
| Space | 49.6% |
| Cardiography | 64% |
| **Average** | **61%** |

Table 12: The average number of component types per architecture for 100 top-performing architectures of each dataset.

| Dataset | Concat | FC | Batchnorm | Dropout | ReLU | Sigmoid | Tanh | Softmax |
|---|---|---|---|---|---|---|---|---|
| Contraceptive | 0.53 | 3.54 | 2.57 | 0.91 | 1.71 | 0.51 | 0.43 | 1 |
| Seismic bumps | 0.47 | 3.53 | 1.75 | 1.62 | 1.68 | 0.64 | 0.48 | 1 |
| Page Blocks | 0.67 | 3.46 | 1.59 | 0.35 | 1.22 | 0.56 | 0.6 | 1 |
| Wind | 0.57 | 3.65 | 2.47 | 0.3 | 1.67 | 0.34 | 0.52 | 1 |
| Puma_32 | 0.55 | 3.56 | 1.84 | 0.94 | 1.6 | 0.46 | 0.57 | 1 |
| CPU_act | 0.65 | 3.95 | 3.23 | 0.2 | 1.91 | 0.3 | 0.63 | 1 |
| Delta elevators | 0.64 | 3.4 | 1.81 | 0.49 | 1.34 | 0.38 | 0.59 | 1 |
| Mammography | 0.69 | 3.6 | 3 | 0.22 | 1.63 | 0.36 | 0.49 | 1 |
| Ailerons | 0.62 | 3.68 | 2.34 | 0.69 | 1.58 | 0.39 | 0.55 | 1 |
| Bank marketing | 0.52 | 3.41 | 2.23 | 1.19 | 1.82 | 0.53 | 0.35 | 1 |
| German Credit | 0.5 | 3.66 | 2.39 | 1.1 | 2 | 0.42 | 0.31 | 1 |
| Space | 0.63 | 3.8 | 3.32 | 0.28 | 2.29 | 0.35 | 0.35 | 1 |
| Cardiography | 0.55 | 3.61 | 3.29 | 0.1 | 2.34 | 0.26 | 0.35 | 1 |

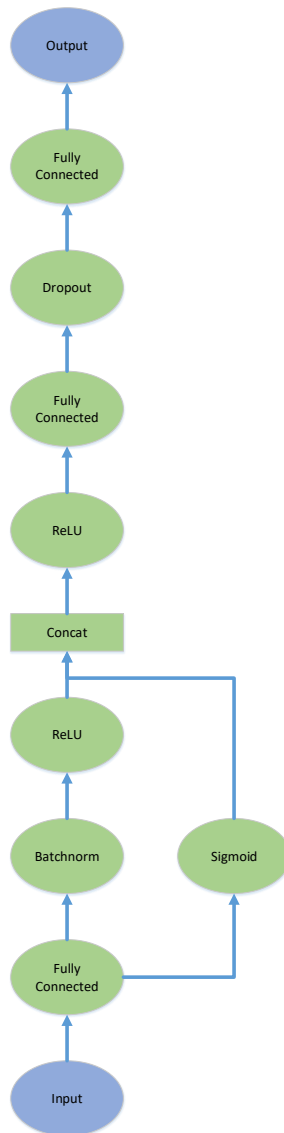

Figure 4: An example of an architecture with parallel layers.

# D  ACCURACY DISTRIBUTION OF THE GENERATED ARCHITECTURES ACROSS THE EVALUATED DATASETS

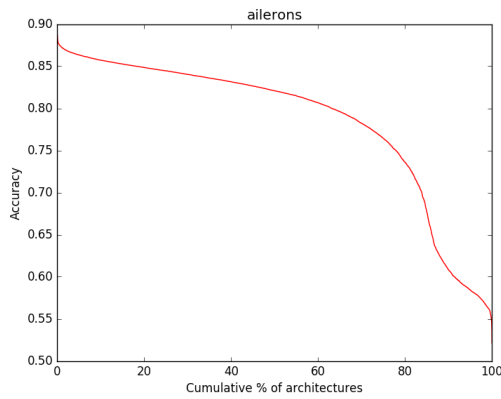

Figure 5: Ailerons

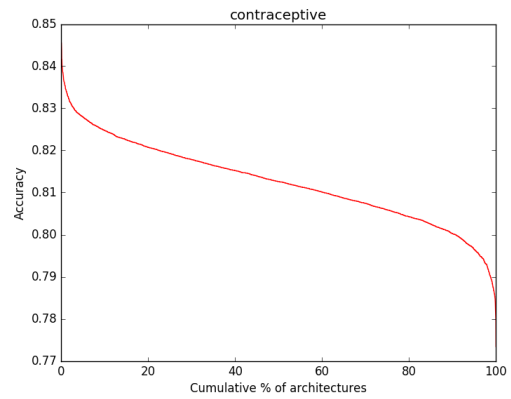

Figure 6: Contraceptive

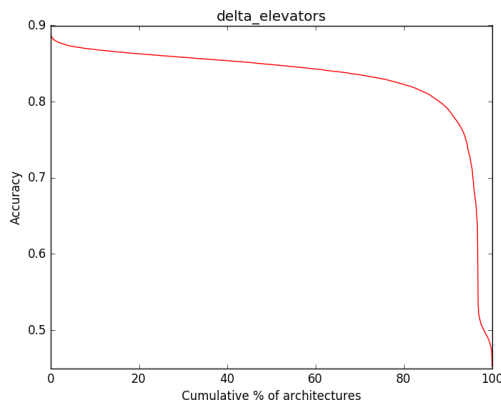

Figure 7: Delta elevators

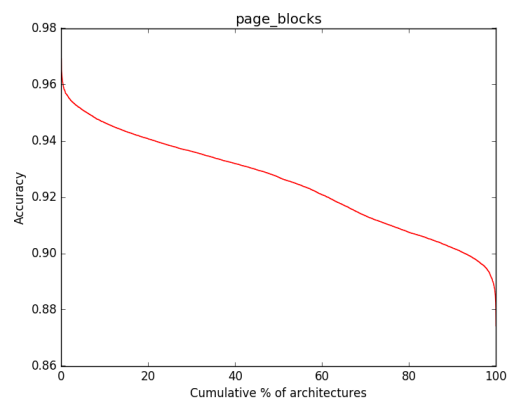

Figure 8: Page blocks

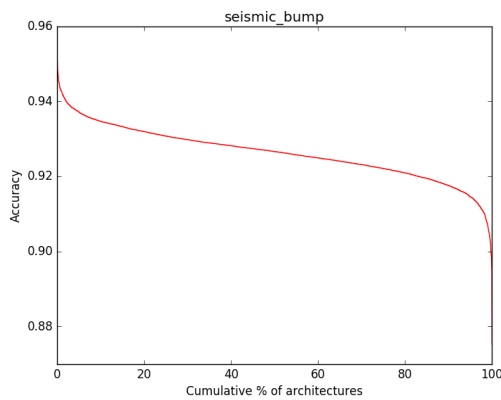

Figure 9: Seismic bumps

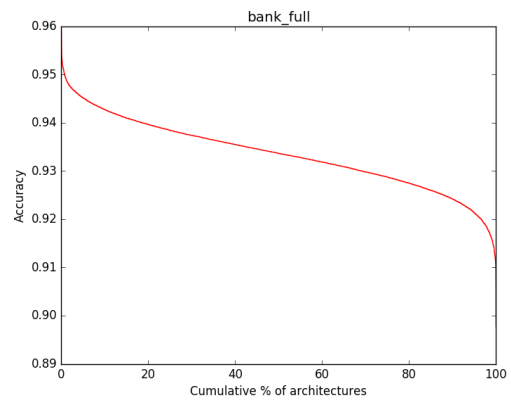

Figure 10: Bank marketing

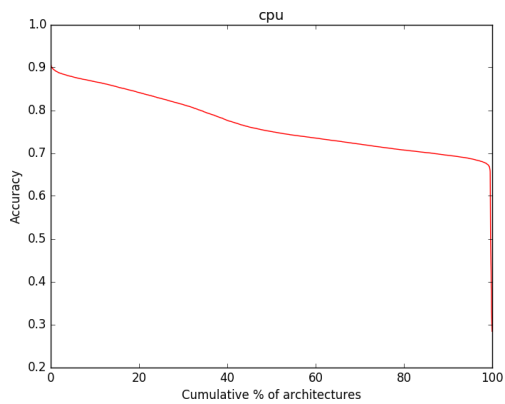

Figure 11: CPU

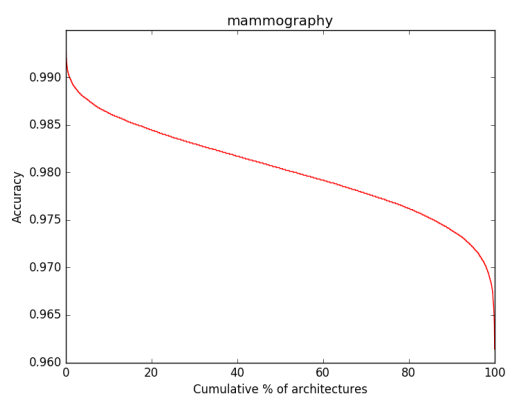

Figure 12: Mammography

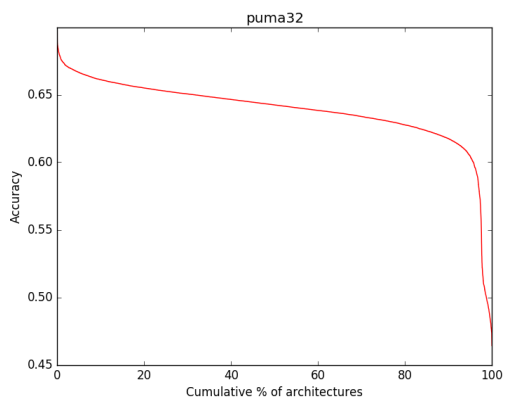

Figure 13: Puma 32NH

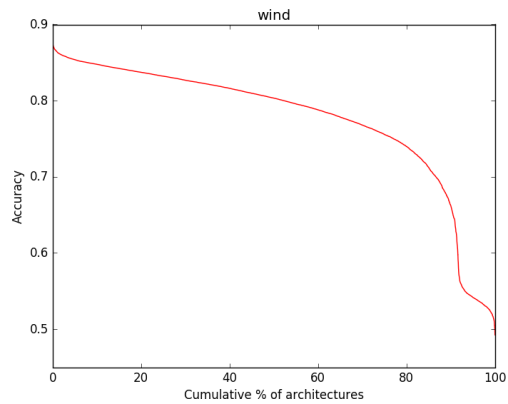

Figure 14: Wind

