# Peer review of "Exploring the Application of Deep Learning for Supervised Learning Problems"

_ICLR 2017 — rejected_

[Official Review · AnonReviewer2 · rating 5 · confidence 4 · 14 Dec 2016]
**An interesting but somewhat underwhelming study**

This paper presents an intriguing study of how one can pose architecture search as a meta learning problem. By collecting features from networks trained on various datasets and training a “ranking classifier” (the actual details of the classifier do not seem to be described in detail) one can potentially infer what a good architecture for a new problem could be by simply running the ranker on the extracted features for a new problem setup.

One notable comment from the paper is that the authors fix some important hyper-parameters for all the networks. I am of the opinion that optimizing the learning rate (and its decay schedule) is actually quite important. I hypothesize that a lot of the conclusions of this paper may change quite a bit if the authors did an actual search over the rates instead. I suspect that instead of training 11k nets, one can train 2k nets with 5 learning rates each and get a much better result that is actually compelling.

I am not convinced that the protocol for generating the various architectures is doing a good job at creating a diversity of architecture (simply because of the max depth of 8 layers and 14 components overall). I suspect that most of these generated architectures are actually almost identical performance-wise and that it’s a waste to train so many of them on so many tasks. Unless the authors are already doing this, they should define a pruning mechanism that filters out nets that are too similar to already existing ones.

The batch normalization experiments in Table 2 seem odd and under-explained. It is also well-known that the optimal learning rates when using batch norm vs. not using batch norm can differ by an order of magnitude so given the fixed learning rate throughout all experiments, I take these results with some grain of salt.

I am not sure we got many insights into the kinds of architectures that ended up being at the top. Either visualizations, or trends (or both), would be great.

This work seems to conflate the study of parallel vs. serial architectures with the study of meta learning, which are somewhat distinct issues. I take issue with the table that compares parallel vs. serial performance (table 2) simply because the right way would be to filter the architectures by the same number of parameters / capacity.

Ultimately the conclusion seems to be that when applying deep nets in a new domain, it is difficult to come up with a good architecture in advance. In that sense, it is hard to see the paper as a constructive result, because it’s conclusions are that while the ranker may do a good job often-times, it’s not that reliable. Thus I am not convinced that this particular result will be of practical use to folks who are intending to use deep nets for a new domain.

[Official Review · AnonReviewer1 · rating 3 · confidence 5 · 15 Dec 2016]
**Interesting first step but not ready for publishing**

This paper aims at attacking the problem of preselecting deep learning model structures for new domains. It reported a series of experiments on various small tasks and feed-forward DNNs. It claims that some ranking algorithm can be learned based on these results to guide the selection of model structures for new domains.

Although the goal is interesting I found their conclusion is neither convincing nor useful in practice for several reasons:

1. They only explored really simple networks (feed-forward DNNs). While this significantly limited the search space, it also limited the value of the experiments. In fact, the best model architecture is highly task (domain) dependent and the type of model (DNN vs CNN vs LSTM) is often much more important than size of the network itself.
2. Their experiments were conduced with some important hyper parameters (e.g., learning rate schedule) fixed. However, it is well known  that learning rate often is the most important hyper parameter during training. Without adjusting these important hyper parameters the conclusion on the best model architecture is not convincing.
3. Their experiments seem to indicate that the training data difference is not important. However, this is unlikely to be true as you would definitely want to use larger models (total number of parameters) when your training set is magnitude larger (i.e., log(datasize) can be an important feature). This is likely because they did not run experiments on large datasets.

In addition, I think the title of the paper does not accurately reflect what the paper is about and should be modified. Also, this paper cited Sainath et al. 2015 as the work that leads to breakthrough in speech recognition. However, the breakthrough in ASR happened much earlier. The first paper with all three key components was published in 2010:

Yu, D., Deng, L. and Dahl, G., 2010, December. Roles of pre-training and fine-tuning in context-dependent DBN-HMMs for real-world speech recognition. In Proc. NIPS Workshop on Deep Learning and Unsupervised Feature Learning.

and the more detailed paper was published in 2012

Dahl, G.E., Yu, D., Deng, L. and Acero, A., 2012. Context-dependent pre-trained deep neural networks for large-vocabulary speech recognition. IEEE Transactions on Audio, Speech, and Language Processing, 20(1), pp.30-42.

As a conclusion, this paper presented some very preliminary result. Although it's interesting it's not ready for publishing.

[Official Review · AnonReviewer3 · rating 4 · confidence 3 · 19 Dec 2016]
**Not convincing**

The topic is very interesting, but the paper is not convincing. Specifically, the experiment part is weak. The study should include datasets that are familiar to the community as well as the ones "that are not often addressed by deep learning". The comparison to other approaches is not comprehensive.

[Final Decision · Program Chairs · 06 Feb 2017]
**ICLR committee final decision**

The reviewers unanimously recommend rejection.